# [Reproducibility Report]
# Double-Hard Debias: Tailoring Word Embeddings for Gender Bias Mitigation

## Reproducibility Summary

**Scope of Reproducibility**

Our goal was to reproduce the original paper's central claim that projecting away the word-frequency direction(s) in word embeddings improves the de-biasing performance of the well-known Hard Debias algorithm. The objectives were to first verify that such word-frequency direction(s) exist and then verify removing these direction(s) decreases bias without significantly affecting the embeddings' utility.

**Methodology**

We were able to use the author's supplied code and raw data as a starting point, though several modifications and additions were inserted into the code to replicate the author's full data pipeline. Specifically, we needed to insert code to export intermediate data files between Jupyter notebooks. In terms of computational resources, we were able to execute all of the code on our local laptop machines after installing specified software dependencies.

**Results**

We were able to reproduce the author's findings that suggest the existence of a word-frequency direction. Specifically, we confirmed that with GloVe embeddings, the second principle component corresponds with the word-frequency direction.

However, we were only able to partially show that projecting away the word-frequency direction improved de-biasing. Our de-biased embeddings contained more bias than the embeddings reported in the original paper. That said, we verified that the Double-Hard de-bias algorithm generally decreases bias when compared with the reproduced Hard de-biased embeddings. Also, the Double-Hard de-biased embeddings preserved semantics, with our scores exactly matching those in the original paper.

**What was easy**

The author's provided much of the code required so re-constructing the data pipeline was not very challenging.

**What was difficult**

The main difficulty involved identifying precisely where our results began to diverge with the authors'. Because there was limited logging in the Jupyter notebook it was difficult to determine why our embeddings were more biased.

**Communication with original authors**

We were in communication with Tianlu, the primary author, on several occasions. Tianlu provided timely responses that confirmed our approach matched the description laid out in the paper. In the end, we were not able to collectively determine the cause behind the divergence in results. However, our findings do reproduce the qualitative finding that a word-frequency direction exists.

# 1 Introduction

In this reproducibility report, we detail our experience reproducing the paper titled "Double-Hard Debias: Tailoring Word Embeddings for Gender Bias Mitigation", which was published at ACL 2020. The paper's main claim is that the existing, well-known word embedding de-biasing technique known as Hard Debias [1] can be improved with an additional pre-processing step that removes the word-frequency direction, hence the naming "Double-Hard Debias". Previous studies have identified that word-embeddings are skewed by the frequency with which words appear in the training data set. Subsequently, the Double-Hard-Debias authors posit that projecting away the word-frequency direction will improve downstream de-baiasing algorithms that identify and remove the gender direction. In their results, the authors demonstrate that their proposed pre-processing technique succeeded in decreasing bias without sacrificing utility.

# 2 Scope of reproducibility

The findings introduced in the previous section can be categorized into three claims presented in the Double-Hard Debias paper:

1. Word-frequency direction(s) can be identified among the word embeddings' principle components.

2. Hard Debias will achieve better performance once the word-frequency direction(s) have been projected away.

3. Double-Hard de-biased embeddings will still perform similarly well as Hard Debiased embeddings on semantic tests.

In reproducing the paper's central claims, we mapped each claim to tables and figures in the original paper. Specifically, claim 1 corresponded to reproducing Figure 2; claim 2 corresponded with Table 3 and Figure 3; and claim 3 corresponded with Tables 2 and 4. At the same time, it is worth mentioning the findings in the paper that were not within the scope of our reproducibility effort. Below, we list the findings that were not within the scope of our replication as well as our reasoning for excluding them:

- We did not reproduce Figure 1 in the original paper, which demonstrates that the gender direction vectors changed significantly after removing the word-frequency direction. We did not reproduce this figure because the code was not provided for the visualization and the findings suggested in this figure can also be uncovered from the classification accuracies.

- The coreference resolution results in Table 1 were also beyond the scope of this report. After consulting with the authors, we discovered that the coreference code exceeded our computing resources. Furthermore, the substantive results can also be shown through the classification accuracies.

- The original paper compares Double-Hard Debias embeddings against not just the Hard Debias embeddings but also a handful of other benchmark embeddings. Because the authors did not generate these embeddings themselves, we did not reproduce the metrics associated with these benchmark embeddings. Instead, we focused on comparing the Double-Hard Debiased embeddings against the Hard Debiased embeddings.

# 3 Methodology

## 3.1 Code and Data

We were able to reconstruct the majority of the data pipeline by forking the authors' Github repository[1]. We needed to execute the Jupyter notebook `GloVe_Debias.ipynb` for claim 1 and to generate the embeddings for claims 2 and 3, which we evaluated with `Glove_Eval.ipynb`. The authors also provided the raw GloVe embeddings data through a download link in the README. Of the various data files provided, we downloaded `vectors.txt`, which are simply the raw GloVe embeddings, and placed this file into the `/data` directory.

In order to successfully execute the pipeline end-to-end however, we had to make several minor changes to the provided code. First, the provided code did not save the intermediate de-biased output from `Glove_Debias.ipynb` to a data file. So, we inserted commands to pickle the de-biased embeddings. Second, the Word Embeddings Benchmark dependency required manual setup during the installation process. In addition to modifying the file paths in the dependency code, we downloaded the analogy metadata text files directly from source as we encountered issues with the auto-download feature[2]. Finally, the author's code did not provide all WEAT scores, so we expanded the tests to provide all metrics.

---

[1] `https://github.com/uvavision/Double-Hard-Debias`
[2] Additional details included in our README

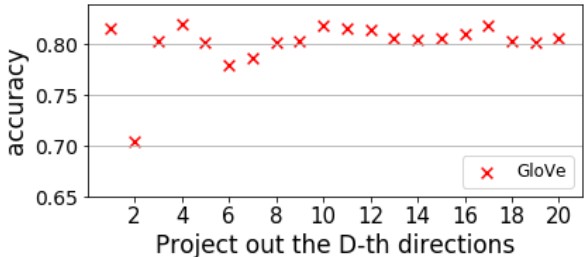

Figure 1: Clustering accuracy after projecting out $D^{\text{th}}$ dominating direction and applying Hard Debias

## 3.2 Computational requirements

The two Jupyter notebooks executed on the GloVe embeddings were computationally inexpensive. We were able to execute each notebook in under 10 minutes on our Macbook Air laptops. However, it should be noted that the authors did report findings based on Word2Vec in their Supplemental Materials. While these results were not within our reproducibility scope, we found that executing the same notebooks on the Word2Vec embeddings was not feasible on our laptops. Furthermore, the most computationally expensive portion of the data pipeline, in our experience, was executing the analogy tests in the evaluation notebook.

# 4 Results

We were able to confirm the existence of a word-frequency direction, and generate Double-Hard de-biased word embeddings using code provided. Though our embeddings did not exactly match those that the authors produced, they performed qualitatively similarly in removing bias and performed well on semantic benchmarks. In this section, we show our results from identifying word-frequency direction among the word embeddings' principle components. Then, we compare the performance of our reproduced Double-Hard de-biased word embeddings against the results reported in the paper. For all experiments, we also reproduced the results the authors reported for raw GloVe embeddings. These results were based on the unmodified, standard GloVe embeddings, so we had a greater expectation that our evaluation metrics would match those in the original paper. In addition, we reproduced the Hard de-biased embeddings to compare the performance of our reproduced Double-Hard de-biased word embeddings to confirm if removing word-frequency direction reduces gender bias. Our results show that Double-Hard de-biased embeddings are overall less biased than Hard de-biased embeddings.

To prove Claim 1, we used GloVe embeddings to see if we could identify a word-frequency distribution among the word embeddings' principle components, which would then lead to better de-biasing results. As shown in Figure 1, we were able to confirm the author's claim that projecting out the $2^{\text{nd}}$ direction leads to the least clustering accuracy, where less accuracy means less bias. Therefore, our results match the author's findings and we were able to have better de-biasing results by projecting out the second principle component.

In our efforts to reproduce the Double-Hard de-biasing results, we generated word embeddings that were more biased than those reported in the paper. The paper evaluated the de-biased word embeddings with two sets of tests: WEAT and classification accuracy. Table 1 contains the WEAT scores reported in the paper as well as score for our reproduced embeddings (green when equal to the original, and red otherwise). When we evaluated the raw GloVe embeddings as well as those we generated with the author's de-biasing code, we were able to match the WEAT scores for both sets of embeddings. However, the gender classification scores for our Double-Hard embeddings were higher than those reported in the paper, as shown in Table 2. Regardless of the sample size, our classification accuracy was at least 10 percent greater than the paper's values, indicating the reproduced embeddings were more biased. This finding is supplemented in Figure 2 which shows the t-SNE clustering plots for our reproduced Double-Hard de-biased embeddings against the plot in the paper. The t-SNE plots confirm that in our embeddings, the male and female words are much more separable.

There are several possible explanations for why our Double-Hard de-biased embeddings are more biased than those described in the paper. Our leading hypothesis is that the authors performed additional PCA projections. In the paper, the authors describe the Double-Hard pre-processing as simply projecting away the second principal component. With the author's code we are able to confirm that the second component is the most optimal component to project away, but the provided code does not show the projection itself.

Table 1: Comparison WEAT Scores Between Original and Reproduced Embeddings

| Embedding | Version | Career ($d$) | Career ($p$) | Math ($d$) | Math ($p$) | Science ($d$) | Science ($p$) |
|---|---|---|---|---|---|---|---|
| GloVe | Original | 1.81 | 0 | 0.55 | 0.14 | 0.88 | 0.04 |
| | Reproduced | 1.81 | 0 | 0.55 | 0.14 | 0.88 | 0.04 |
| Hard | Original | 1.55 | $2e^{-4}$ | 0.07 | 0.44 | 0.16 | 0.62 |
| | Reproduced | 1.53 | $2e^{-4}$ | 0.10 | 0.57 | 0.21 | 0.66 |
| Double-Hard | Original | 1.53 | $2e^{-4}$ | 0.09 | 0.57 | 0.15 | 0.61 |
| | Reproduced | 1.53 | $2e^{-4}$ | 0.09 | 0.57 | 0.15 | 0.61 |

Table 2: Comparison of Gender Classification Scores for Top-K Gendered Words

| Embedding | Version | Top 100 | Top 500 | Top 1000 |
|---|---|---|---|---|
| GloVe | Original | 100 | 100 | 100 |
| | Reproduced | 100 | 100 | 100 |
| Hard | Original | 59.0 | 62.1 | 68.1 |
| | Reproduced | 66.5 | 77.4 | 81.5 |
| Double-Hard | Original | 51.5 | 55.5 | 59.5 |
| | Reproduced | 66.5 | 74.1 | 70.4 |

Furthermore, we used the author's code to reproduce the Hard de-biased word embeddings and evaluated it against our reproduced Double-Hard de–biased word embeddings. Although the reproduced Hard de-biased embeddings did not provide the same WEAT scores or gender classification scores as the ones from the original paper, we were able to prove that Double-Hard de-biased embeddings were generally less biased than Hard de-biased embeddings. For both gender classification scores for top-500 and top-1000 gendered words, the Double-Hard was less biased than Hard as shown in Table 2. Overall, the reproduced Double-Hard embeddings did better in the WEAT tests with lower scores meaning having less bias than Hard embeddings shown in Table 1. One interesting point is that the original Hard de-biased embeddings had lower gender classification scores meaning less bias than the reproduced version. The discrepancies that are seen between the original reported gender classification scores and the reproduced scores can also point to how the discrepancies exist even before removing the word-frequency direction in Double-Hard de-bias algorithm. Although the evaluation scores did not match exactly with the original reported scores, we were able to prove that Double Hard de-biased embeddings perform better than Hard de-biased embeddings.

Interestingly, even though the Double-Hard de-biased embeddings that we generated were more biased than those the authors created, we were able to exactly match the analogy and categorization benchmark test scores. Table 3 includes the original and reproduced benchmark scores for the GloVe, Hard de-biased, and Double-Hard de-biased embeddings. Given the fact that the Double-Hard scores match, we can conclude that our embeddings perform just as well on standard word-embedding applications as the authors' embeddings, validating claim 3. Of note, we found discrepancies in the evaluation metrics for the raw GloVe and Hard de-biased embeddings. Our best explanation for the GloVe embeddings is that the authors uploaded a newer version of GloVe vectors to their data repository after publishing the paper earlier this year; on the other hand, we were not able to explain the differences for the Hard De-bias embeddings, but these differences do not affect our stated claims.

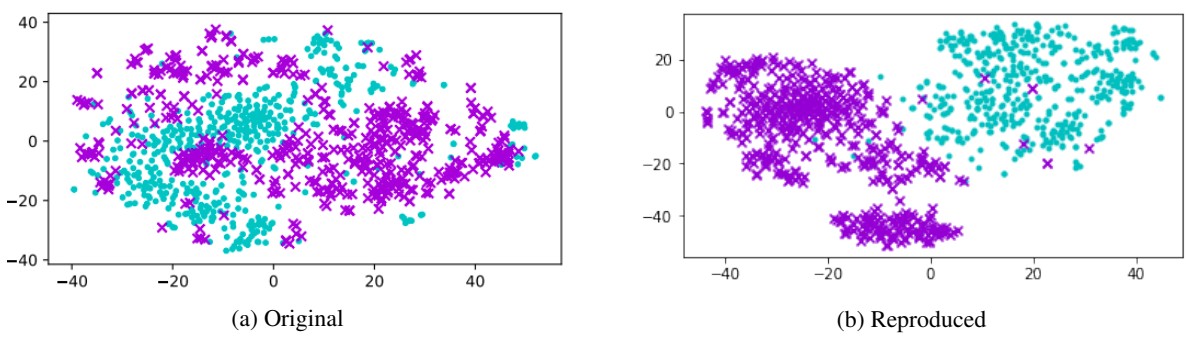

(a) Original          (b) Reproduced

Figure 2: Comparing original vs reproduced embedding t-SNE clustering using Double-Hard

Table 3: Performance of Original vs. Reproduced Embeddings on Benchmarks

| Embedding | Version | Sem | Syn | Total | MSR | AP | ESSLI | Battig | BLESS |
|---|---|---|---|---|---|---|---|---|---|
| GloVe | Original | 80.5 | 62.8 | 70.8 | 54.2 | 55.6 | 72.7 | 51.2 | 81 |
| | Reproduced | 80.5 | 62.8 | 70.8 | 54.2 | 58.9 | 72.7 | 49.6 | 81 |
| Hard | Original | 80.3 | 62.5 | 70.6 | 54.0 | 62.3 | 79.5 | 50.0 | 84.5 |
| | Reproduced | 80.3 | 62.5 | 70.6 | 54.0 | 57.1 | 72.7 | 46.6 | 78.0 |
| Double-Hard | Original | 80.9 | 61.6 | 70.4 | 53.8 | 59.6 | 72.7 | 46.7 | 79.5 |
| | Reproduced | 80.9 | 61.6 | 70.4 | 53.8 | 59.6 | 72.7 | 46.7 | 79.5 |

## 5 Discussion

We were able to verify claims 1 and 3 in our original reproducibility roadmap while the experiments for claim 2 did not match precisely. For claim 2, we did not generate the exact same numerical values but we were able to replicate the qualitative finding that Double-Hard de-biased embeddings are less biased than Hard de-biased embeddings. Even though we were not able to match all of the results for claim 2, overall, replicating the results in this paper was made much simpler due to the code and documentation that the authors provided. All of the results were contained within two Jupyter notebooks and the scripts were not computationally expensive. Given further resources, we would aim to reproduce the findings that we excluded from the scope of this report.

### 5.1 What was easy

The author's provided much of the code required so replicating the data pipeline was not very challenging. It was helpful that most of the code was written in a modular, organized, and efficient way. For this reason, it was not difficult to make simple changes to the code. It was also helpful that the authors specified which dependencies to install.

### 5.2 What was difficult

The main difficulty involved identifying precisely where our results began to diverge with the authors'. In the end we were able to reproduce the analogy metrics but not the classification accuracy values. Because there was limited logging in the Jupyter notebook it was difficult to determine why our embeddings were more biased.

### 5.3 Communication with original authors

We were in communication with Tianlu, the primary author, on several occasions. Tianlu provided timely responses that confirmed our approach matched the description laid out in the paper. Tianlu was able to advise us that the coreference results would require significant computational investments, which directed our efforts. In terms of debugging the slight differences we noticed in the classification accuracies we were not able to collectively determine the cause behind the divergence in results.

Based on our discussion, we believe that the differences may be due to two reasons. First, as noted in the GloVe_Eval.ipynb notebook, the random seeds included in the published code may have differed from those used while writing the original paper. The seeds are needed, in particular, to ensure reproducible clustering with the KMeans module in scikit-learn. Second it is possible that the list of most biased words – needed for Table 2 of this report – differed between our execution and the authors'. The two most probable causes for this difference are that the authors uploaded a slightly newer version of the GloVe embeddings when releasing the data and that the authors utilized alternative metrics to rank gender bias, as commented in the evaluation notebook. However, our findings do reproduce the qualitative finding that a word-frequency direction exists.

## References

[1] Tolga Bolukbasi, Kai-Wei Chang, James Zou, Venkatesh Saligrama, and Adam Kalai. Man is to computer programmer as woman is to homemaker? debiasing word embeddings. In *Proceedings of the 30th International Conference on Neural Information Processing Systems*, NIPS'16, page 4356–4364, Red Hook, NY, USA, 2016. Curran Associates Inc.

