# OpenReview forum: "[Reproducibility Report] Double-Hard Debias: Tailoring Word Embeddings for Gender Bias Mitigation"
_ML_Reproducibility_Challenge/2020 — Reject_

### Official Review · AnonReviewer2 · 2021-03-01
**Reproduction of Double Hard Debias**

**Rating:** 8
**Confidence:** 3

**Review:**

The reproducibility report of the Double Hard Debias:Tailoring Word Embeddings for Gender
Bias Mitigation (Wang et al) clearly identifies the three claims made by the author and sets the motivation to reproduce these claims.  The methodology used is also clearly outlined and there was communication with the original authors to ensure that experiments were performed without deviation from the original. The results obtained on reproduction match two of the three claims made by the original authors. The reproducibility reports also attempts to find reasons why one experiment did not have identical results reported on the original papers but these hypotheses were not validated.

**Familiar With The Original Paper:**

I have read the original paper

**Reproducibility Summary:**

Report has summary

---

### Official Review · AnonReviewer3 · 2021-03-10
**Reproduced specific and characteristic results in the original paper to an extent.**

**Rating:** 6
**Confidence:** 4

**Review:**

Authors have presented a well-organized reproducibility report. Scope of reproducibility was categorized into clear and independent claims, which were representative of and covers the extent and novelty of the original source paper being reproduced.

To verify the original results, claims were further mapped to the corresponding qualitative and quantitative findings of the source work, thereby making the subsequent sections easy to follow. Not all the results in the source paper has been attempted to be replicated, however, the reasons to exclude them have also been explicitly mentioned (either computational constraint or the results substantiating the claims which are also corroborated by other presented results).

Authors have used the source code made public by the authors of the original paper. While the codes were modified to fit in a pipeline to run them, as mentioned by the authors, they also expanded the set of WEAT scores to cover all the metrics reported in the source paper. Communication with original authors were also established, particularly regarding the deviations of results from the source paper. Although the differences could not be resolved, discussions on reasons for the same is also briefly included along with the observation that the results qualitatively holds, albeit to a lesser extent. For example, substantiating claim 2, the gender classification scores reported in Table 2 of the reproducibility report for top K gendered words were observed to be less when compared with Glove embeddings but were still greater than the reported values in the original work. Similarly, to substantiate claim 3, original and reproduced performances on word analogy and concept categorization benchmark datasets are presented in Table 3 of the reproducibility report and small differences are observed on concept categorization dataset.

Although it is mentioned as well as noted that the results specific to the claims in the scope of reproducibility will be reported, it is observed that Figure 2 of the original work pertaining to claim 1 in the reproducibility report regarding identification of word-frequency direction(s) among word embeddings' principal components is not included in the report.

Of further note is the omission of reproducibility of scores and results pertaining to Hard (Debiased) Glove embeddings in tabular results i.e. Table 1,2 and 3. This is in contrast with the outlined scope of reproducibility where it was explicitly mentioned that the current work's focus is on comparing the Double-Hard Debiased embeddings against the Hard Debiased embeddings. Thus, this doesn't necessarily reproduces the results qualitatively, particularly when there are significant differences. For example, claim 2 in the reproducibility report is regarding reduced gender-bias in moving from Hard Debias to Double-Hard Debias and the same is also claimed in the original paper. In the original paper, this is demonstrated through clustering accuracy/gender classification scores. In the reproducibility report, while Double-Hard Debias results have been reported and are found to be worse when compared to source paper, reporting the Hard Debias results would have substantiated or refuted the gender bias reduction in qualitative sense (although still differing) when moving towards Double-Hard Debiasing.

Furthermore, the results, associated discussions particularly when calling out observed discrepancies/differences needs to be highlighted properly. For example in Table 3 in the reproducibility report, while differences in AP and Battig categorization tests have been highlighted and reported, differences for other tests, some of which are even more pronounced, are still marked as matching.

Finally, discussion on results and difficulty in reproducibility can be made more thorough, specifically, set of choices that could be made and lead to difference in results. For example, for results pertaining to Table 3 where classification scores on top K gendered words are reported, the original work mentions identifying such words based on cosine similarity with gender direction in the original GloVe embedding space. Similarly, it is mentioned that random seeds could also be one of the factors causing differences in the results. Discussion and documentation on the gender direction used, range of random seeds used (given that experiments in the scope were not too computationally expensive) are further needed to cover the extent of reproducibility.

**Familiar With The Original Paper:**

I have read the original paper

**Reproducibility Summary:**

Report has summary

---

### Official Review · AnonReviewer1 · 2021-03-15

[review text omitted: it was posted to a different submission]

---

### Decision · Program_Chairs · 2021-03-31

**Decision:**

Reject

**Comment:**

Overall reviews and/or the paper content not good enough for the AC to recommend to the journal.